# Link-centric analysis of variation by demographics in mobile phone communication patterns

**Mikaela Irene D. Fudolig**[1]*, **Kunal Bhattacharya**[2,3], **Daniel Monsivais**[3], **Hang-Hyun Jo**[1,4,3], **Kimmo Kaski**[3,5]

**1** Asia Pacific Center for Theoretical Physics, Pohang, Republic of Korea, **2** Department of Industrial Engineering and Management, Aalto University School of Science, Espoo, Finland, **3** Department of Computer Science, Aalto University School of Science, Espoo, Finland, **4** Department of Physics, Pohang University of Science and Technology, Pohang, Republic of Korea, **5** The Alan Turing Institute, London, England, United Kingdom

* mikaela.fudolig@apctp.org

**Data Availability Statement:** All the relevant data behind this paper has been deposited in a public data depository: https://doi.org/10.6084/m9.figshare.9165746.

## Abstract

We present a link-centric approach to study variation in the mobile phone communication patterns of individuals. Unlike most previous research on call detail records that focused on the variation of phone usage across individual users, we examine how the calling and texting patterns obtained from call detail records vary among pairs of users and how these patterns are affected by the nature of relationships between users. To demonstrate this link-centric perspective, we extract factors that contribute to the variation in the mobile phone communication patterns and predict demographics-related quantities for pairs of users. The time of day and the channel of communication (calls or texts) are found to explain most of the variance among pairs that frequently call each other. Furthermore, we find that this variation can be used to predict the relationship between the pairs of users, as inferred from their age and gender, as well as the age of the younger user in a pair. From the classifier performance across different age and gender groups as well as the inherent class overlap suggested by the estimate of the bounds of the Bayes error, we gain insights into the similarity and differences of communication patterns across different relationships.

## Introduction

The availability of population-level communication data has made it possible to study human interaction patterns on very large scales [1–4]. In particular, the utilization of mobile phone usage data along with anonymized demographic information of users has proven to be extremely effective in understanding the age and gender dependence [5–12], as well as the innate behavioral traits [13–15]. Additionally, the general network-based research approach has been furthered by the inclusion of machine learning to exploit the differences in social interaction found in different age and gender types and to predict demographic information of individuals. For example, de Montjoye *et al.* [16] predicted personality traits of users based

**Funding:** D.M., K.B., and K.K. acknowledge support from EU HORIZON 2020 INFRAIA-1-2014-2015 program project (SoBigData) No. 654024. K. K. also acknowledges the Rutherford Foundation Visiting Fellowship at The Alan Turing Institute, UK. H.-H.J. acknowledges financial support by Basic Science Research Program through the National Research Foundation of Korea (NRF) grant funded by the Ministry of Education (NRF-2018R1D1A1A09081919). The funders had no role in study design, data collection and analysis, decision to publish, or preparation of the manuscript.

**Competing interests:** We have read the journal's policy and the authors of this manuscript have the following competing interests: Hang-Hyun Jo is currently serving on the editorial board of PLOS ONE. This does not alter our adherence to PLOS ONE policies on sharing data and materials.

on their call detail records using an SVM classifier. Similar frameworks using standard machine learning methods have been used to predict gender and age [17–20]. Different other techniques including factor graph models [21], homophily-based methods [22, 23] and neural networks [24, 25] have also been adopted. These studies have mainly relied on the egocentric perspective, working on the implicit assumption that an individual's traits determine his or her calling and texting patterns. Thus, when using an egocentric approach in predicting individual user information, all calls and texts made by the target individual are aggregated over all its neighbors, so the links in the target individual's egocentric network are not differentiated from each other.

While the communication patterns have been shown to vary with age and gender at the individual level as discussed above, the communication patterns between two individuals have also been found to vary with the relationship between them [26, 27]. However, while an egocentric perspective works well in studying variation at the individual level, it cannot, by design, provide much insight into variation among pairs. We hope to expand knowledge in this area by focusing on the links rather than the nodes in the mobile phone communication network, analyzing each link independently from others instead of aggregating them over the egocentric network. This link-centric perspective has not been explored much in relation to mobile phone data [28, 29], especially in prediction tasks. In particular, only Herrera *et al.* [24] have used isolated link information as well as the demographic information of one user in the link to predict the age and gender of the other user; thus, predictions are made without the need for full information on the ego's calling and texting activities, which is required in egocentric approaches. While Herrera *et al.*'s aim was to predict individual user information, we are interested in the connection between the relationship between users and the patterns of communication between them. To understand this, we first have to look at how to represent communication patterns and how to characterize the relationship between users. For the former, we can represent the raw call detail records as a set of relevant features, which is a rather standard approach. The latter, however, is more challenging and requires more thought.

Relationships, unlike age and gender, cannot be categorically confirmed from call detail records, and determining the relationship between any pair of users from the mobile phone data relies on inference. In particular, unless otherwise specified, we mean such inference from the relationship and not the true relationship whenever we refer to "relationship" in this paper from this point onwards. David-Barrett *et al.* [30] used age and gender differences to infer the relationships between the pairs of users, such as mother-daughter, father-son, spouses, etc. On the other hand, Fudolig *et al.* [31] extended this approach to add call frequency ranking information to infer different relationships even among pairs with similar demographics. As we shall see later, although we only perform prediction tasks for information with ground truth (age and gender), we also analyze our results in the context of these inferred relationships based on the age, gender, and ranking information.

Indeed, taking a link-centric approach gives rise to a large number of research questions that involve relationships between the pairs of users, and the paradigm can also be extended to go beyond such dyads, tackling questions related to the relationships among members in groups [9]. To demonstrate its use, however, we narrow down our interest to mutual top-rank pairs, where each individual in the pair is the most frequently called alter of each other. As calling frequency has been shown to be related to social closeness [32], these pairs are likely to correspond to very close relationships such as romantic, platonic, or familial ones. Thus, in mutual top-rank pairs, each user in a pair is a very, if not the most, important alter in the other user's social network, and our previous research [31] has shown significant differences in the behavior between mutual top-rank pairs and those which are not.

Our aim is to study the variation in mobile communication patterns among mutual top-rank pairs and how this variation relates to the relationships between users as inferred from their age and gender. We begin by exploring the aspects of the calling and texting patterns that contribute the most to the variance in the data. Then, to investigate the feasibility of using a link-centric approach to predict user information, we perform two prediction tasks by using standard machine learning techniques on call detail records: one inferring the relationship between users, and the other inferring the age of the younger user in a pair. For the former case, we specifically look at identifying opposite-gender peers among mutual top-rank pairs, a relationship which we find interesting because they form the majority of mutual top-rank pairs and are also most plausibly characterized as romantic pairs, as discussed in our previous study [31]. Although we have shown there how mutual top-rank opposite-gender peers differ from other types of peer relationships, we have yet to see whether these opposite-gender peers are actually distinguishable from non-peer relationships (e.g., parent-child relationships) among mutual top-rank pairs. From the performance of the classifiers, both in aggregate as well as for different types of age and gender combinations, we obtain deeper insights into how communication patterns vary across relationships.

## Methodology

### The dataset

We use anonymized call detail records (CDRs) from January to July 2007 from a single service provider in a European country. The call detail records include the time and duration of all the calls and text messages (also referred to as "SMS"; marked with zero duration) made to and from company service subscribers. However, although the calls made from non-subscribers to subscribers are recorded, their durations are not. Further, the age, gender and billing post code are included only for company service subscribers.

For every subscriber, we rank its alters, both subscribers and non-subscribers, based on the total number of calls between each pair. We also require that each ego-alter link must have at least one call for five out of the seven months in the data, ensuring regularity in order to filter out transactional calls. We then select only those pairs who are mutually top-ranked, i.e., each user in the pair is the rank-1 alter of the other by call frequency. This results in a total of 256,117 mutual top-rank pairs with available user age and gender information.

We group pairs based on the age difference between the users. We use census data to create three categories that likely correspond to different types of relationships: (a) less than 20 years, (b) 20–39 years, and (c) at least 40 years. Pairs with an age difference of less than 20 years are considered as "peers", while the remaining two categories are similar to the age differences in "parent-child" and "grandparent-child" relationships as inferred from the census data [33]. Further, among peers, we differentiate between *same-gender* and *opposite-gender* pairs. Although the true relationship between the users cannot be obtained from the user metadata in the call detail records, the fact that these users are mutually top-rank suggests that they are in close relationships that are likely familial, platonic, or romantic in nature.

It is entirely possible that the relationships as given by the age and gender difference also change with the age of the users: for example, spouses raising young kids may have different mobile communication patterns compared to older spouses [29, 30]. We thus categorize the pairs by the age bracket of the younger user: <18, 18–28, 29–45, 46–55, 56–79 and ≥80. Some of these categorizations were also used in Fudolig *et al.* [31] and were chosen to correspond to the possible life stage of a particular user as inferred from census data [33]. The age range 18–28 corresponds to "young adulthood" (Y), where the users are probably unmarried; 29–45 is considered "middle adulthood" (M), where users are likely to be starting families; 46–55 is

considered "late adulthood" (L), where users' children have started to become teenagers and adults; while 56–79 is considered "old adulthood" (O). Although rarer than the other categories, some users are also below 18 and above 79, which we consider "teenage years" and "very old adulthood".

Note that although we have employed the above categorizations to infer different types of relationships, we only perform prediction tasks for response variables that are explicitly available in the user metadata (i.e., age and gender). We however will later explore how the performance of our models differ for the different groups we have presented above.

## Generating the features

Since our goal is to examine the variation in calling and texting patterns among pairs, we limit ourselves to information that can be obtained from the call detail records of two users in a pair. These includes the calls and texts that the users make to each other, as well as the local network structure in the mobile phone communication network.

We extract features from the dataset as summarized in Table 1. Most features were segmented into Mondays to Thursdays (weekdays) and Fridays to Sundays (weekends), as

**Table 1. Features obtained from the call detail records.** *These features were log-transformed. †Log transformation was applied for late-night quantities involving the numbers of calls and call durations. ‡Log transformation was applied for every feature.

| *Quantity of interest* | *Day/time division* | *Final quantities used as features* |
|---|---|---|
| Weekly<br>(a) number of calls,<br>(b) call duration and<br>(c) number of texts | Weekday/weekend daytime/evening/late-night | Mean*, median*, std*, min*, max*, skewness, kurtosis (taken over the entire observation period) |
| Total<br>(a) number of calls,<br>(b) call duration and<br>(c) number of texts<br>for the entire observation period | Weekday/weekend daytime/evening/late-night | Fraction of [weekday, weekend] [calls, call duration, texts] in daytime/evening/late-night† |
| (a) Number of days with at least one call and<br>(b) number of days with at least one text<br>for the entire observation period | Weekday/weekend daytime/evening/late-night | Number of days with at least one [call, text] in each division of time‡ |
| Reciprocity for<br>(a) number of calls,<br>(b) call duration and<br>(c) number of texts<br>for the entire observation period | N/A | \|in−out\| ÷ (in+out) for each quantity |
| (a) Interevent time for calls and<br>(b) interevent time for texts<br>for the entire observation period | N/A | Mean*, median*, std*, min*, max*, skewness*, kurtosis* |
| Number of common contacts | N/A | Number of common contacts within top 5 most called alters |
| | | Number of common contacts among all alters |

well as daytime (7am–4pm), evening (5pm–10pm), and late-night (11pm–6am) segments. These segmentations were based on the periods of high and low activity in the aggregate calling behavior of the population [34]. However, some features were calculated over the whole 7-month period, such as the inter-event time distribution as well as the number of common contacts. For the number of common contacts, we include two measurements, one for the total number of common contacts and the other for the number of common contacts among the top 5 ranked alters, which are considered the closest alters of the user [35, 36]. In total, 175 features were obtained for each pair. Log transformations were used for features with long-tailed distributions. All features were then standardized to mean 0 and standard deviation 1.

## Data analysis

We use principal components analysis (PCA) to examine the variation of the calling and texting patterns across the pairs of users. The loadings are then used to interpret these components qualitatively.

To demonstrate whether such variation in the data can give us demographics-related quantities for one or both users in a pair, we perform two binary classification tasks: one to identify which among the mutual top-rank pairs are opposite-gender peers (age difference of less than 20 years), and another to find the age group of the younger user in each pair ($<35$ and $\geq 35$). A test set of $n_{\text{test}} = 20000$ was obtained from the full dataset of mutual top-rank pairs, while the training sets of varying sizes were drawn from the remaining portion. This ensures that the test set does not intersect with any in the training set. Then balanced training sets are made by taking an equal number of samples corresponding to each value of the response variable. Thus, the training sets used depend on the classification task considered, although the rest of the procedure is identical for identifying the opposite-gender peers and predicting the age group of the younger user in a pair.

Feature selection is performed by logistic regression (LR) and Linear SVM (LSVM) with L1 penalty, with coefficients less than $10^{-5}$ omitted, similar to the methodology employed in Ref. [19]. Each prediction model is fitted via 5-fold cross-validation optimized for accuracy on three different sets of features: the original set, the one from feature selection by LR, and the one from feature selection by LSVM. The following prediction models were used: logistic regression (LR), linear SVM (LSVM), SVM with RBF kernel (SVM), random forest (RF), and $k$-nearest neighbors (KNN). The tuned hyper-parameters were the regularization parameters for LR, LSVM, and SVM; the number of trees for RF; and the number of nearest neighbors for KNN. Each model is fitted using five seeds, and the final prediction is taken to be the mode of the predicted values of the response variable for all the seeds. The performance of each combination of feature selection method and prediction model is then reported. The Python library `scikit-learn` (version 0.20.0) was used to implement all the models.

As we are using a limited set of machine learning models, one can ask if the classifier performance is due to the behavior of the classifier in relation to the data or to the overlap in the class distributions themselves. If there is significant overlap in the class distributions in the feature space, we do not expect any classifier to separate the two classes well. To address this question, we compare the performance of our classifier with the *Bayes error*, $E_{bayes}$, the minimum possible error rate due to the overlap of class distributions [37]. The maximum possible accuracy is thus given by $1 - E_{bayes}$. A higher Bayes error, which corresponds to a lower accuracy, indicates more overlap in the class distributions in the feature space. Thus, if a particular classifier has an error close to the Bayes error, we can deduce that the results of the classifier are closely related to the inherent overlap in the class distributions.

The Bayes error can be computed exactly when the *a priori* class probability $P(c_i)$ of each class $i$ and class likelihoods $p(x|c_i)$ are known, where $x$ is a vector in the feature space. The *Bayes classifier* assigns the vector $x$ to the class with the highest posterior probability $p(c_i|x)$, and the error associated with this classifier is the Bayes error,

$$E_{\text{bayes}} = 1 - \sum_i \int_{C_i} P(c_i)p(x|c_i)dx \qquad (1)$$

where the integral is over the region $C_i$ where class $i$ has the highest posterior [37].

Although the Bayes error is hard to determine exactly since the required probabilities are generally unknown, its upper and lower bounds can be estimated. To estimate the lower and upper bounds of the Bayes error, and therefore the upper and lower bounds of the accuracy of the Bayes classifier, we use the estimate obtained from the error rate of the 1-nearest neighbor (1-NN) classifier with a sufficiently large training set, $E_{NN}$ [37, 38]:

$$\frac{1 - \sqrt{1 - 2E_{NN}}}{2} \leq E_{bayes} \leq E_{NN} \qquad (2)$$

Note that although Eq 2 uses a the 1-NN classifier to estimate the bounds of the Bayes error, the Bayes error itself is independent of the classifier used for a particular classification problem.

## Results

### Variation in the communication patterns across different pairs

PCA was performed on a randomly selected subset ($n = 20000$) of the mutual top-rank pairs dataset. The first principal component (PC) explains 28% of the variance, while the second and third PCs explain 10% and 7% of the variance, respectively. The fourth and succeeding PCs all explain a portion of the variance 5% and below. The scree plot (Fig 1) shows an elbow at around $n_{\text{comp}} = 5$, with 54% of the variance explained.

The results of the PCA can be interpreted within the framework of factor analysis. The factor analysis assumes that the observed features are generated by some latent factors, which are in turn interpreted by looking at the corresponding factor loadings. Features with absolute factor loadings above some cutoff are considered to represent one factor. The PCA estimates these loadings with the eigenvectors scaled by the variance explained, and these loadings can then be rotated for ease of interpretation; details are given in Ref. [39].

With this perspective, we perform rotation on the loadings of the top $n_{\text{comp}} = 5$ PCs and use the cutoff of 0.4 for the absolute loadings; oblimin and varimax rotations give similar results. We find that different times of the day correspond to different factors, while a mix of weekday or weekend-related features are found in each factor. Call- and text-related quantities are also in separate factors. These indicate that most of the variation in the data can be accounted for mainly by the time at which the calls or texts are made and the type of communication channel used (call or text) (Table 2). The number of common neighbors as well as the reciprocity between users are not found to contribute significantly to the top 5 factors.

### Identifying opposite-gender peers

We now look at the performance of the different classifiers used in predicting whether users in a pair are opposite-gender peers or not. Of all the classifiers, SVM is found to perform the best, yielding an accuracy of 67% for $n_{\text{train}} = 20000$, with LSVM, LR, and RF giving only slightly lower accuracy at 66%. KNN is found to be the worst prediction model used, giving accuracies

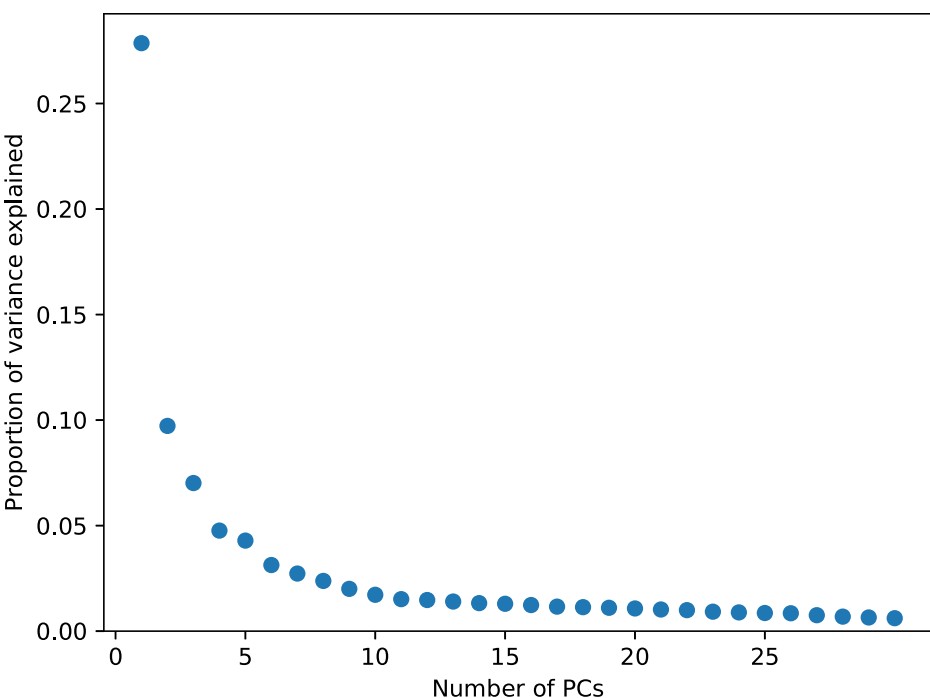

**Fig 1. Scree plot obtained from PCA.**

around 60% (Fig 2). Performing feature selection does not significantly affect the accuracy, and often resulted in slightly worse performance than using the full set of features. The accuracy is higher than those obtained from a random classifier and a majority classifier (classify all as opposite-gender peers) which would give 50% and 61% test accuracy, respectively. Using the SVM on the full set of features also gives a precision of 0.74, a true positive rate of 0.70 and true negative rate of 0.61.

Beyond $n_{train}$ = 10000, the accuracies begin to plateau, and the gains in accuracy by increasing the training set size (from $n_{train}$ = 20000 to 30000) are outweighed by the amount of computational resources required for these traditional machine learning models. Further, we also estimate the upper bound of the maximum possible accuracy using Eq 2 to be 68%. This indicates that there is considerable irreducible error due to class distribution overlap, and that our best classifier already has near-optimal performance given the hand-engineered features provided. It may be possible to employ end-to-end deep learning to make use of the raw calling and texting patterns directly to improve performance, which we consider as a promising direction for future study.

Table 3 gives the performance of the SVM (with $n_{train}$ = 20000) for each type of relationship defined by the age and gender difference, focusing on those types that comprise at least 1% of the test set (this covers 97.7% of the test set). It is found that the classifier correctly identifies a large majority of opposite-gender peers in the 18–28 (Y) (86% accuracy) and 29–45 (M) (75% accuracy) age groups, but fails even below the random baseline for those in the 46–55 (L) and 56–79 (O) age groups. Conversely, it correctly identifies many same-gender peers in the older age groups (L and O), but misclassifies many in the younger age groups (Y and M) as opposite-gender peers. We also note that while the prediction accuracy decreases for opposite-gender peers as one goes from younger to older age groups, it increases for the same-gender peers.

**Table 2. Factors accounting for the top 5 principal components in the dataset with the corresponding top 5 features assigned to each and their absolute loadings.** Loadings were obtained by performing an oblimin rotation on the loadings obtained from the PCA.

| Factor | Features with top 5 absolute loadings | Loadings |
|---|---|---|
| Daytime calls (7am–4pm) | Mean number of daytime calls in a week on weekdays | 0.90 |
| | Number of days with daytime calls on weekdays | 0.89 |
| | Stdev of number of daytime calls in a week on weekdays | 0.82 |
| | Median of number of daytime calls in a week on weekdays | 0.82 |
| | Maximum number of daytime calls in a week on weekdays | 0.80 |
| Evening calls (5pm–10pm) | Number of days with evening calls on weekdays | 0.82 |
| | Number of days with evening calls on weekends | 0.79 |
| | Mean number of evening calls in a week on weekdays | 0.79 |
| | Mean number of evening calls in a week on weekends | 0.77 |
| | Median evening call duration in a week on weekdays | 0.76 |
| Late-night calls (11pm–4am) | Fraction of late-night calls (frequency) on weekdays | 0.89 |
| | Fraction of late-night calls (duration) on weekdays | 0.89 |
| | Fraction of late-night calls (frequency) on weekends | 0.88 |
| | Fraction of late-night calls (duration) on weekends | 0.88 |
| | Maximum number of late-night calls in a week on weekdays | 0.84 |
| Texts | Mean number of daytime texts in a week on weekdays | 0.92 |
| | Mean number of daytime texts in a week on weekends | 0.92 |
| | Mean number of evening texts in a week on weekdays | 0.91 |
| | Number of days with daytime texts on weekdays | 0.90 |
| | Number of days with daytime texts on weekends | 0.90 |
| Texts | Median interevent time for texts | 0.60 |
| | Mean interevent time for texts | 0.58 |
| | Minimum interevent time for texts | 0.55 |
| | Skewness of daytime texts in a week on weekdays | 0.54 |
| | Maximum interevent time for texts | 0.53 |

Similar behavior is obtained for all the other classifiers, indicating that this result is reflective of the distribution of the samples in the feature space and not of the type of classifier used.

Thus we see that a classifier trained to differentiate between opposite-gender peers and other relationships varies in performance depending on the age range. Note that since the accuracy computed from the Bayes error estimate is close to the accuracy of the best performing classifier that we found, it is highly likely that this age dependence is not an artifact of the model used but is rather related to the overlap in the distributions of opposite-gender peers and other types of pairs in the feature space. The bulk of the opposite-gender peers in the training data belong to the age groups Y and M, so a good classifier is expected to classify these age groups reasonably well. Specifically, for the SVM, one can imagine the separating hyper-surface in the feature space to separate opposite-gender peers in the age groups Y and M from those in other relationships. If the opposite-gender peers in the age groups L and O have similar behavior to those in the age groups Y and M, we expect them to lie on the same side of the hyper-surface even if they are less represented in the training set than those in the age groups Y and M. However, as this is not observed, it suggests that, in general, the opposite-gender peers in age groups L and O may have different calling and texting patterns from their younger counterparts. On the other hand, more same-gender peers in the Y and M groups are misclassified than their L and O counterparts despite the Y and M groups having more representation in the train set.

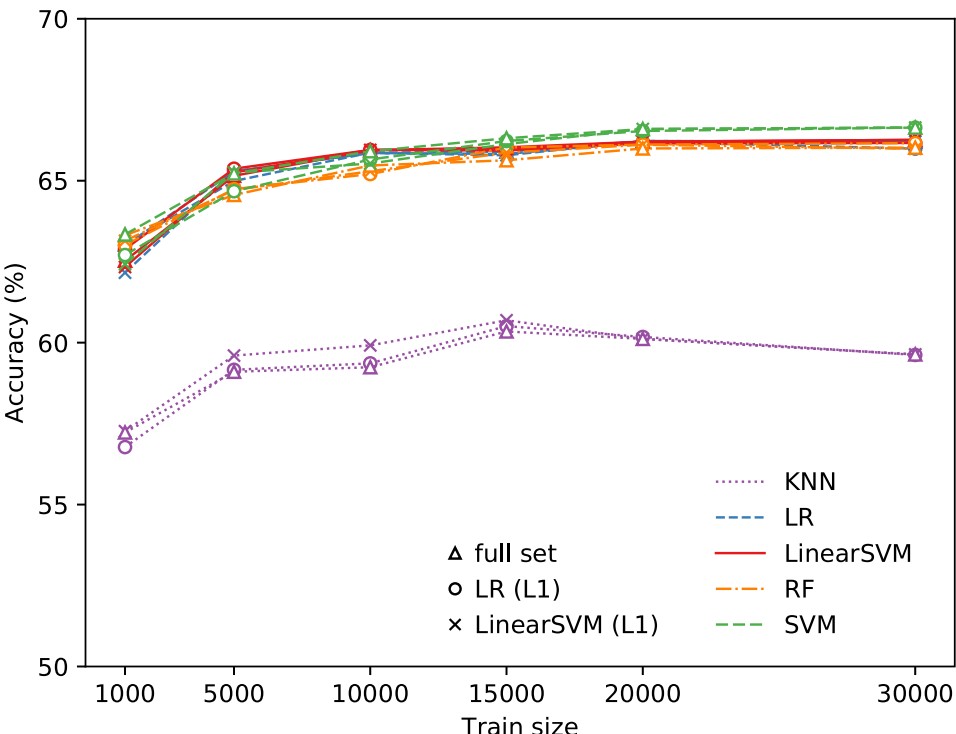

**Fig 2. Test accuracy for the different training sizes and prediction models in identifying opposite-gender peers.**
The symbols ○ and × indicate the accuracies obtained using logistic regression and linear SVM, respectively, for feature
selection, while the triangles (△) denote accuracies obtained using the full set of features.

We can further expound on our observations discussed above by looking at the obtained
posterior probabilities. These give us an idea on how confident the classifier is about its predictions. Although the SVM itself does not produce the probabilities that a particular sample
belongs to a class, these can be approximated by additionally training a sigmoid function that
maps the SVM decision function to probabilities [40].

Fig 3a shows, for peer groups, the relative frequencies of the probabilities that users in a particular pair are opposite-gender peers. The peaks of the relative frequencies for the Y and M
opposite-gender peers indicate that the model is confident that these are indeed opposite-gender peers, while most opposite-gender peers in the O age group are assigned by the classifier to
the other class confidently but incorrectly. On the other hand, there is more uncertainty
among opposite-gender peers in the L group as can be seen from the flatter curve near $p = 0.5$.
These observations about the posterior probabilities reinforce the age dependence we found in
the performance of the SVM among opposite-gender peers.

For same-gender peers, the SVM assigns the L and O groups confidently and correctly, but
makes errors in the Y and M groups. Notably, we see an almost bimodal distribution for the M
same-gender peers, with the two peaks almost at equal height and far enough from $p = 0.5$.
Majority of the Y same-gender peers, on the other hand, are confidently and incorrectly classified as opposite-gender peers. Thus, among same-gender peers, we also find that the classifier
performance depends on age.

To see if the classifier would perform better among peers if the age dependence were
removed, we performed the training and testing only on a particular peer age group. For example, we used a balanced training set consisting solely of peers in the O age group and evaluated
the classifier's performance by looking also at a test set of peers in the O age group. We then

**Table 3. Accuracy of best performing models ($n_{train}$ = 20000) in identifying opposite-gender peers (OGP) and predicting the age of the younger user (YUA) for each relationship as inferred from a pair's age and gender difference.** For peers, + indicates same-gender pairs, while − indicates opposite-gender pairs. Y corresponds to pairs where the younger user is in the age range 18–28; M, 29–45; L, 46–55; O, 56–79. Peers are indicated by "peers", while parent-child relationships are denoted by "child". Note that in the latter case, it is the age group of the child that is given, while the parent is at least 20 years older. For the YUA prediction, we also include information about the composition and accuracy found for the subgroups of those in age group M corresponding to cases where the YUA is below and at least 35 years old.

| Relationship code | Accuracy (%) | | % of test set | % of train set | |
|---|---|---|---|---|---|
| | OGP | YUA | | OGP | YUA |
| −Y peers | 86.1 | 88.8 | 13.8 | 11.6 | 13.3 |
| +Y peers | 38.0 | 85.1 | 4.3 | 5.6 | 4.2 |
| −M peers | 74.8 | 66.0 | 36.4 | 30.1 | 36.6 |
| +M peers | 50.2 | 66.8 | 10.6 | 12.9 | 10.7 |
| −L peers | 40.0 | 82.9 | 7.1 | 5.4 | 6.9 |
| +L peers | 69.3 | 77.5 | 3.2 | 4.3 | 3.3 |
| −O peers | 23.1 | 91.5 | 3.3 | 2.8 | 3.0 |
| +O peers | 80.4 | 88.2 | 1.8 | 2.5 | 2.2 |
| Y child | 56.6 | 63.5 | 6.3 | 7.9 | 6.0 |
| M child | 72.5 | 63.4 | 9.6 | 12.2 | 10.0 |
| L child | 84.0 | 90.9 | 1.3 | 1.7 | 1.4 |
| −M peers ($< 35$) | | 69.1 | 16.6 | | 16.4 |
| −M peers ($\geq 35$) | | 63.4 | 19.8 | | 20.2 |
| +M peers ($< 35$) | | 63.3 | 4.5 | | 4.0 |
| +M peers ($\geq 35$) | | 69.4 | 6.1 | | 6.6 |
| M child ($< 35$) | | 41.4 | 4.4 | | 4.3 |
| M child ($\geq 35$) | | 81.9 | 5.3 | | 5.7 |

compared it to the accuracy obtained using the classifier trained on the full set (i.e., not restricted to pairs of the same age group) for every age group as given in Table 3. We find that training on an age-restricted set gives a higher accuracy for the L and O peers but a lower accuracy for the Y and M peers (Table 4). By breaking down each age group into opposite-gender and same-gender peers and obtaining the accuracy per subgroup (which corresponds to the true positive rate and true negative rate, respectively), we find that this is because using the age-restricted balanced training sets also distributes the error across the two different classes more evenly. Note though that identifying the gender of users using the age as an input may not have much practical sense: if the age of a user is given, it is highly likely that the gender is also available. However, the above analysis reveals the extent to which the models could be influenced by the age groups Y and M that dominate the full training set.

As for the pairs inferred to have parent-child relationships, the classifier distinguishes them from opposite-gender peers reasonably well. Nevertheless, we also see that the classifier performance is better for pairs where the younger user is in older age groups. From the posterior probabilities in Fig 3b, we see that for parent-child pairs where the child is in the Y age group, the curve is flatter, with a smaller peak at $p > 0.5$. We have attempted to resolve this bimodality by separating the pairs into smaller subgroups, taking the gender of the users into account (i.e. mother-daughter, mother-son, father-daughter, father-son pairs). However, the behaviors for all subgroups were found to exhibit bimodality as well, indicating that the gender difference of the parent-child pairs is not enough to explain this discrepancy.

## Identifying the age of younger user in a pair

In attempting to classify the opposite-gender peers versus others among mutual top-rank pairs, we have found that the classifier's performance depends on the age of the users. We

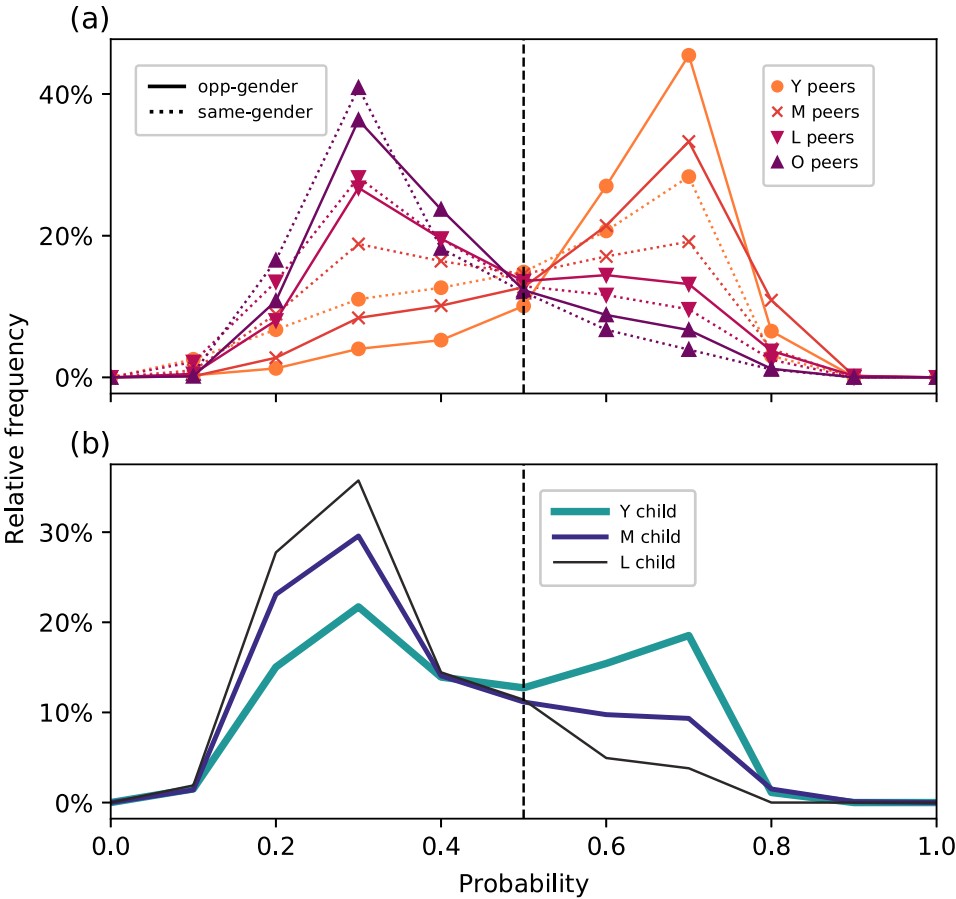

**Fig 3. Relative frequencies for each relationship category among (a) peers and (b) parent-child pairs showing the probabilities that users in a particular pair are opposite-gender peers.** The dashed vertical line shows $p = 0.5$, the case where the classifier assigns equal probabilities to whether users in a pair are opposite-gender peers or not.

hypothesized that this may translate to an age dependence in the communication patterns as well. To investigate this further, we look at how well we can predict the age of the younger user based on the calling and texting patterns of a pair. Note that although we are predicting demographic information of an individual, the approach is still link-centric. Previous egocentric approaches use aggregate calling and texting patterns per individual as predictors, while we use the communication patterns between two users.

Regression is performed on the dataset to see if the exact age of the younger user can be predicted from the 175 features extracted from the calling and texting patterns. The ridge, lasso,

**Table 4. Accuracy (in %) in predicting if a pair of peers is opposite-gender for peers in different age groups using the full and age-restricted training sets.** The OGP columns give the accuracy among opposite-gender peers, the SGP columns among same-gender peers, and the OGP+SGP columns among all peers in the given age group.

| Peer age group | OGP+SGP | | OGP only | | SGP only | |
|---|---|---|---|---|---|---|
| | Full | Age-restricted | Full | Age-restricted | Full | Age-restricted |
| Y | 74.6 | 70.2 | 86.1 | 75.9 | 38.0 | 52.6 |
| M | 69.3 | 64.5 | 74.8 | 67.6 | 50.2 | 53.8 |
| L | 49.2 | 58.2 | 40.0 | 57.0 | 69.3 | 60.5 |
| O | 43.3 | 58.4 | 23.1 | 60.4 | 80.4 | 54.6 |

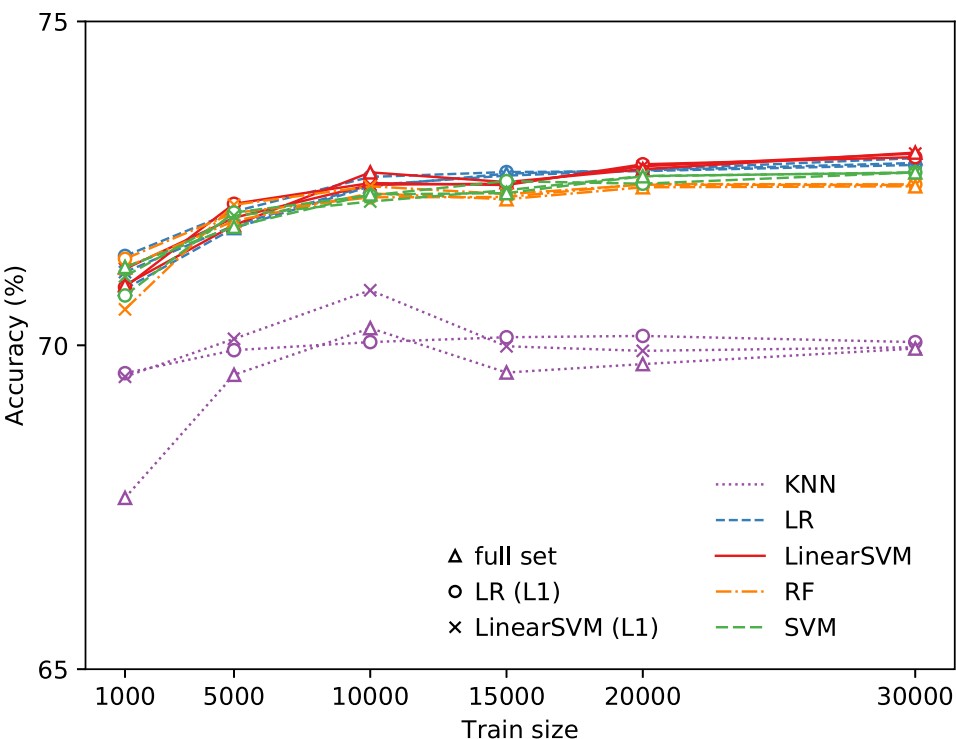

**Fig 4. Test accuracy for the different training sizes and prediction models in identifying whether the younger user of a pair is less than 35 years of age or not.** The symbols ○ and × indicate the accuracies obtained using logistic regression and linear SVM, respectively, for feature selection, while the triangles (△) denote accuracies obtained using the full set of features.

and random forest regression are all performed with the regularization coefficients tuned via 5-fold cross-validation. Of these models, a maximum $R^2$ value of 0.30 was obtained even with a train size of $n_{\text{train}} = 20000$; no significant improvement over a train size of $n_{\text{train}} = 5000$ was observed. This indicates that the models using the hand-engineered features from the call detail records between pairs are not enough to predict the exact numerical age.

We then try to predict the age group of the users based on whether they are "young" or "old", setting our cutoff to be 35 years of age. The cutoff value was chosen so that most points near the decision boundary will be confined to only one age group and so as not to give a dataset highly skewed towards either the "young" or "old" groups. This yields a dataset of mutual top-rank pairs where 52% of pairs have younger users less than 35 years old.

All the classifiers are found to perform better than the baseline of 50% for a random classifier and 52% for a majority classifier (Fig 4). Even the KNN, which has the worst performance among all the models selected, performs much higher than the baseline. The LSVM turns out to be the best classifier with the accuracy of 73% at $n_{\text{train}} = 20000$, with SVM, LR, and RF giving very close results. The feature selection has a minimal effect on the performance. The performance of LSVM, SVM, LR and RF also plateau early on; beyond $n_{\text{train}} = 10000$, only a slight increase in the accuracy is obtained with larger training set sizes. The LSVM gives a precision of 0.75, a true positive rate of 0.71 and a true negative rate of 0.74. Overall, these results indicate that a linear classifier would work well for such a prediction task. The estimated upper bound for the maximum possible accuracy by Eq 2 is 76%, indicating that the best classifier is performing close to optimal.

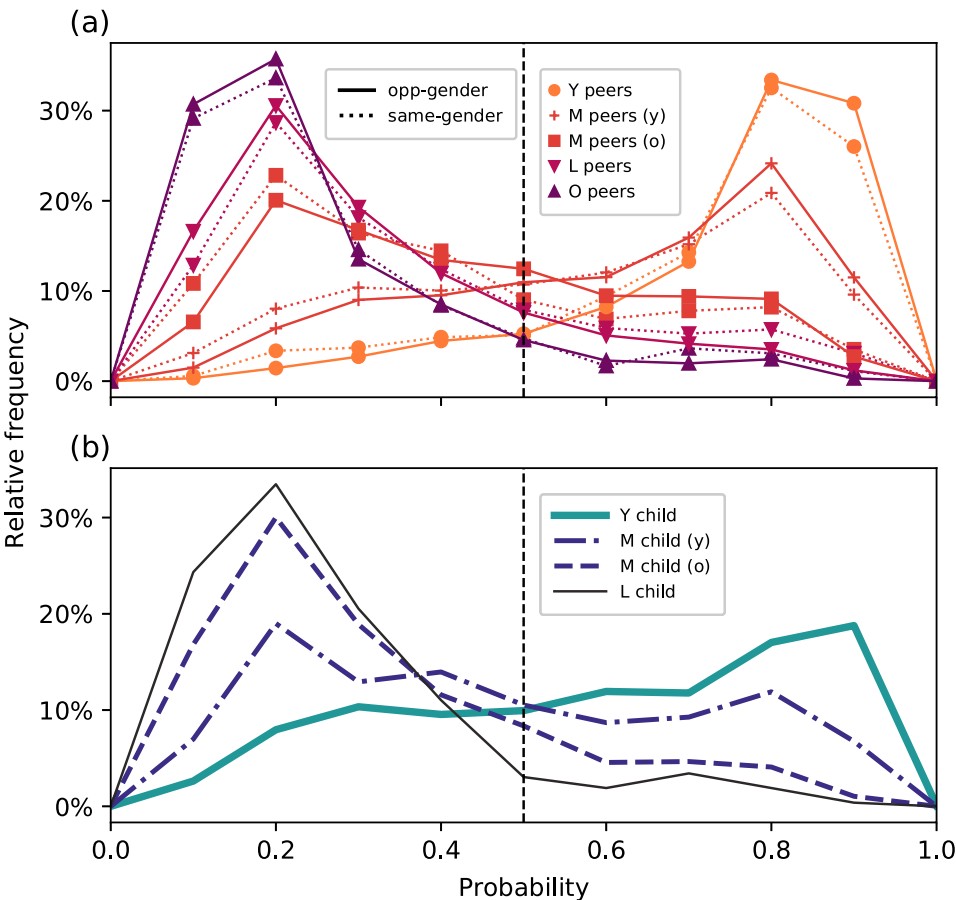

**Fig 5. Relative frequencies for each relationship category among (a) peers and (b) parent-child pairs showing the probabilities that the younger user of a particular pair is <35 years old.** The "y" and "o" suffix refer to the subgroups of pairs in the M age group that are below and at least 35 years old, respectively. The dashed vertical line shows $p = 0.5$, the case where the classifier assigns equal probabilities to whether the younger user's age is below or at least 35.

Although our Bayes error estimate indicates that there is significant overlap between the two classes, the model has good accuracy for most types of relationships (Table 3). As expected, since the cutoff is in the age group M, all relationships in this category suffer from poorer performance compared to others but it is still well above the baseline. This indicates that much of the overlap in class distributions is near the age cutoff, since our results indicate that the pairs much younger and much older than 35 years of age are far enough in the feature space to be separated by a hyperplane. To further understand the performance in the age group M, we separate each category in this age group into two subgroups, those with the younger user less than 35 years old and at least 35 years old. For peers, we find that the accuracy is between 60–70%, above the baseline but still below those in the other age groups. On the other hand, we observe an accuracy below the baseline (41%) for parent-child pairs where the younger user is below 35; for those where the younger user is at least 35 years old, the accuracy is 82%. These can also be observed in the relative frequencies plot for the posterior probabilities shown in Fig 5.

These results support our earlier observation that peer groups are more separable by their age rather than their gender difference, indicating that there is more variation in communication patterns in the feature space among different age groups than among opposite-gender and same-gender peers. However, for parent-child pairs, the classifier performs well only for

those where the child is in the older age groups (i.e. O, L, and M≥35) and poorly in the younger age groups. Since the classifier performance is markedly different for peers and parent-child pairs, these imply that what may distinguish older from younger peers is different from what characterizes older and younger parent-child pairs.

## Discussion

To our knowledge, this study is the first of its kind to look at prediction tasks in large-scale mobile phone data with a specific focus on relationships between service users as inferred from their age and gender. Most research involving the prediction of demographic information from mobile phone data employ an egocentric perspective, with the implicit assumption that an ego's characteristics determine his or her communication patterns. Here, we have explored this issue from a link-centric perspective, looking at how communication patterns change across various pairs of users. Although Herrera *et al.* [24] have previously explored isolated link calling and texting patterns to predict the user's age and gender, the dependence of the classifier on the relationships between the pair of users was not studied. Further, their approach had a strong egocentric component, as it aimed to identify the age of a particular target user; notably, one method that was explored involved using the age and gender of the the target user as an input in the model. In contrast, we focus more on the relationship between the users and leverage the observation that communication patterns vary with relationship. This approach gives us a more complete picture of the variation in mobile phone communication patterns, complementing the studies done from an egocentric point of view.

We have found that the variation in the calling and texting patterns among mutual top-rank pairs can be mostly explained by the type of communication channel (calls or texts) used, as well as their timings, while the day of the week, reciprocity between individuals and local network information are found to be less relevant. In order to explore whether this variation can be used to separate some pairs from others, we performed two classification tasks: one to identify opposite-gender peers, and the other to predict the age of the younger user of the pair.

Predicting opposite-gender peers yielded a reasonable accuracy of 67%. This is close to the estimated upper bound of the Bayes classifier accuracy, indicating near-optimal performance, as well as a significant overlap of opposite-gender peers and other types of pairs in the feature space. Further, we note that although SVM with an RBF kernel yields the best performance, linear classifiers such as linear SVM and logistic regression offer less computational cost at only slightly lower accuracy.

By performing a post-hoc analysis across different types of relationships, we found that the classifier performance depends on the age of the users. Opposite-gender peers are correctly identified among younger peers but heavily and confidently misclassified among older peers, with the prediction accuracy decreasing as one goes from younger to older age groups. Conversely, the same-gender peers are mostly properly identified among older peers, but not among younger peers, with the prediction accuracy increasing as one goes from younger to older age groups. On the other hand, the parent-child pairs are classified mostly accurately, but the performance is poorer among younger age groups. Specifically, the pairs where the child is in the Y age range seem to be composed of two subgroups, with one behaving more like opposite-gender peers than the other. We attempted to resolve this apparent bimodality by decomposing the pairs further by their gender difference (e.g., mother-daughter, father-daughter, etc.) but found no difference in the classifier performance among these groups. These results are consistent with the overlap suggested by our estimates for the bounds of the Bayes error.

This age dependence of the classification accuracy is consistent with earlier studies that revealed age- and gender-dependent factors influencing human sociality observed in mobile phone data [12, 41–43]. As mentioned earlier, the vast majority (around 84%) of opposite-gender peers in the training set are in the younger age groups Y and M. Thus, since we intentionally do not use age and gender as an input to the model, the model is expected to associate the label "opposite-gender peers" with the calling and texting behavior found in opposite-gender peers in the age groups Y and M. These age groups also correspond to the age range where spouses or partners, which are the most likely categorization of these top-ranked opposite-gender peers, are reproductively active. In this period, they have the calling and texting behavior characterized by an overall stronger mutual focus [41, 44] reflected by high frequency and regularity of communication [31]. However, for older spouses or partners, a dilution in the communication intensity between them is expected as their social focus shifts to their children [41]. This shift from spouse to children is reflected in the lower accuracy among younger parent-child pairs, as the parent in these pairs are most likely in the L and O groups. The finding that the opposite-gender peers in the L and O groups are also highly misclassified, indicating a different behavior from their younger counterparts, is also consistent with this shift in social focus. On the other hand, we expect that the same-gender peers are not constrained by reproduction costs and family maintenance and will rather be influenced by a broad mixture of strategies that govern the dynamics in the egocentric networks of individuals [42].

The age dependence of the users' communication patterns can also be seen from the success of predicting information about the age of the younger user in a pair. Although standard regression techniques performed poorly in predicting the specific age, similar to what was observed in egocentric prediction studies [19], we obtained good results in predicting whether the younger user is below or at least 35 years of age. We find that although our estimate of the bounds of the Bayes error still indicates significant overlap in the feature space, this overlap is mostly limited to points near the age cutoff (age group M), as the accuracy for this age group is lower than in others.

The Linear SVM, the best-performing classifier, has an accuracy of 73% even with a training set size of $n = 10000$. This is close to the upper bound estimate of the Bayes classifier accuracy, indicating near-optimal performance. This classifier yields very good accuracy for Y, L, and O peers at 77% accuracy and above, indicating good separability in the feature space. Thus, the communication patterns as described by the 175 features used are sufficiently different for these age groups to separate them into those with the younger user below or at least 35 years old. M peers, on the other hand, have lower but almost equal performance for those below 35 and at least 35 years of age, which is expected, since those near the boundary (for example, younger user of age 34 vs. younger user of age 36) may have similar behavior but will belong to different categories.

Although the same classifier performs extremely well for older parent-child pairs (younger user above 35 years of age), it has poorer performance for younger parent-child pairs, especially for those where the child is in the M group but below 35 years of age. As this behavior is different from those found among peers, this suggests that the age dependence among peer communication patterns as given in the feature space is not the same as that among the parent-child pairs.

The poor accuracy among the parent-child pairs where the child is in the M group but less than 35 years of age may be due to the variations inherent in this group. The age group of the younger user (29-34 year old) likely corresponds to the very early stages of forming a family, where the younger user may have young infants or toddlers, or no children at all. It is possible that the calling patterns between the younger users and their parents may change significantly after the younger users have children of their own. Aside from this, there may also

be differences between parent-child pairs that co-reside and those that do not. Further investigation into the location data as well as egocentric communication patterns in this group may shed light on this issue [29, 30].

One should also note that all our results are based on the representation of communication patterns in the feature space given by the 175 hand-engineered features. It is possible that extracting other features from the call detail records will yield different results. This does not negate our findings, but offers other perspectives on the aspects of communication patterns that we may not have considered in our feature space.

## Conclusion

In summary, we have shown by a link-centric analysis that mobile phone communication patterns between users do vary with the users' age and gender difference. The age and gender difference can in turn be used to infer a plausible relationship between the users. Further, we have seen that this variation allows us to separate some pairs from others, and consequently, allowing us to see which types of pairs are more similar and which are more different from each other in the feature space.

It should be noted, however, that although we predict user information, this user information is obtained *for each pair*; e.g., we could predict the age of the younger user, but we could not determine which of the two users in a pair is younger. Thus, although a link-centric approach works well when focusing on relationships, an egocentric approach is more straightforward if user information is needed at the individual level. Recent egocentric studies on user demographic prediction show good results, with reported accuracies as high as around 75-85% for gender prediction and 60-65% for age group prediction depending on the dataset used [19, 20, 25].

However, since what we predicted in this study are link-based quantities (relationships in terms of age and gender difference), we cannot directly compare our results with those of egocentric approaches which predict individual information (individual age and gender). In practice, one could use individual information to predict the age and gender of individuals, and then use these to obtain the age and gender difference and infer the relationship between two users. However, as this approach aggregates the communication patterns of an individual over all its neighbors, comparing the communication patterns across different relationships, as we have done, will not be possible. Thus, the link-centric approach is not meant to replace egocentric approaches but to complement them in exploring the pairwise nature of communication patterns.

Aside from having an advantage in studying communication patterns across different relationships, link-centric analyses can be used to predict individual information in cases where egocentric analyses cannot. One drawback of egocentric approaches is that complete information on the ego's call and text logs is needed to make a prediction. In contrast, we find that our link-centric approaches may require only knowledge of the records between the ego and only one of its alters, as network metrics (which require full information on the ego's behaviors) do not appear to affect the variation in pairs as much as the other parameters do. Similar to Herrera *et al.*'s method [24], one could use a link-centric approach to predict information about the pair and then use information about the alter to infer information about the ego. This can be applied to, for example, non-subscribers, whose calls and texts are unknown except for those made with company subscribers. A link-centric analysis, perhaps a modified form of the one we did in this paper, can be performed to make inferences about the non-subscribers. Further, our analysis can also be expanded to examine relationships among groups of individuals as well.

Although calling and texting are increasingly getting replaced by app-based communication in smartphones and communication via social media, the problem of prediction of user demographics from device usage data can still be considered relevant. Aside from its academic value in studying human behavior, demographic information is also useful not just in the fields of advertising and marketing but also in social development and public health. Jahani *et al.*, for example, discussed how call detail records can be used to identify women among prepaid users (with no registered demographic information) and send text messages regarding reproductive and child care, or how call detail records can serve as a cheaper proxy to census data for estimating the socioeconomic conditions of certain groups [19]. These are also true for app-based communication, where growing concerns over privacy which could restrict the access to user information. In this regard, the approaches such as ours which being link-centric, basing prediction on limited amounts of data and using features that are mostly time-stamps of events and their counts rather than their explicit content [45], could be effective approaches for modeling and prediction of user attributes and behavior.

## Acknowledgments

We acknowledge the computational resources provided by the Aalto Science-IT project.

## Author Contributions

**Conceptualization:** Mikaela Irene D. Fudolig, Kunal Bhattacharya, Daniel Monsivais, Hang-Hyun Jo, Kimmo Kaski.

**Data curation:** Mikaela Irene D. Fudolig, Daniel Monsivais.

**Formal analysis:** Mikaela Irene D. Fudolig.

**Investigation:** Mikaela Irene D. Fudolig.

**Methodology:** Mikaela Irene D. Fudolig.

**Software:** Mikaela Irene D. Fudolig.

**Supervision:** Hang-Hyun Jo, Kimmo Kaski.

**Visualization:** Mikaela Irene D. Fudolig.

**Writing – original draft:** Mikaela Irene D. Fudolig.

**Writing – review & editing:** Mikaela Irene D. Fudolig, Kunal Bhattacharya, Daniel Monsivais, Hang-Hyun Jo, Kimmo Kaski.

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
