## [Decision Letter · Decision Letter 0]

15 Oct 2019

PONE-D-19-21537

Link-centric analysis of variation by demographics in mobile phone communication patterns

PLOS ONE

Dear Dr. Fudolig,

Thank you for submitting your manuscript to PLOS ONE. After careful consideration, we feel that it has merit but does not fully meet PLOS ONE’s publication criteria as it currently stands. Therefore, we invite you to submit a revised version of the manuscript that addresses the points raised during the review process.

We would appreciate receiving your revised manuscript by Nov 29 2019 11:59PM. To enhance the reproducibility of your results, we recommend that if applicable you deposit your laboratory protocols in protocols.io, where a protocol can be assigned its own identifier (DOI) such that it can be cited independently in the future. For instructions see: http://journals.plos.org/plosone/s/submission-guidelines#loc-laboratory-protocols

We look forward to receiving your revised manuscript.

Kind regards,

Ginestra Bianconi

Academic Editor

PLOS ONE

Journal Requirements:

2) We note that you have stated that you will provide repository information for your data at acceptance. Should your manuscript be accepted for publication, we will hold it until you provide the relevant accession numbers or DOIs necessary to access your data. If you wish to make changes to your Data Availability statement, please describe these changes in your cover letter and we will update your Data Availability statement to reflect the information you provide.

3) Thank you for stating the following in the Competing Interests section:

[I have read the journal's policy and the authors of this manuscript have the following

competing interests: Hang-Hyun Jo is currently serving on the editorial board of PLOS

ONE.].

Reviewers' comments:

Reviewer's Responses to Questions

**Comments to the Author**

1. Is the manuscript technically sound, and do the data support the conclusions?

Reviewer #1: Partly

2. Has the statistical analysis been performed appropriately and rigorously? 

Reviewer #1: Yes

3. Have the authors made all data underlying the findings in their manuscript fully available?

Reviewer #1: No

4. Is the manuscript presented in an intelligible fashion and written in standard English?

Reviewer #1: Yes

5. Review Comments to the Author

Reviewer #1: The authors studied a dataset of mobile phone communications (calls/texts) essentially performing two prediction tasks 1) opposite gender pairs 2) age of younger user in the pair. To this end they proposed a link-centric (i.e. focus is on the pair) rather the classic ego-centric (i.e. focus on the ego) approach. The methodology stems from standard machine learning. Topic is of interest as well as the results. Before publication however some revisions are in order:

The main assumption is that differences in prediction performance across classes are associated to differences in communications patterns between such classes. It is unclear to me however to what extent this is actually proven. At the end of the day the “truth” used here comes from the outcome prediction of a quite limited set of machine learning models. In other words, how can we be sure that such differences are not introduced (even in part) by non-trivial interactions between the models and the samples? This is a key issue of machine learning approaches. In these cases, an independent theory/expectation should come to the rescue. The authors hint in several parts that some observations fit with sociological expectations and previous work however they do not provide enough details in this sense limiting their discussion with few words and citations.

Null models might be very telling. What would happen to the predictions in case the pairs would be shuffled? I am thinking about a shuffling done in the raw data (i.e. single calls) not in the aggregated metrics. The shuffling should remove the various patterns. Thus, would this imply prediction tasks more balanced across type of pairs?

The key element of novelty, which has been nicely stressed, is the focus on links rather than egos. However, it is unclear what is the success of approaches based on the ego. Are they better in predicting gender of pairs? This can be done predicting the gender of connected groups, I am guessing. What about age? As the authors wrote in the conclusion “the problem of prediction of user demographics from device usage data can still be considered relevant” thus it is key to understand the actual benefit of using a link centric approach is in this context. In order words, does using this approach helps? If yes, how much? Clearly the paper is not focusing 100% of improving a prediction task, but since it sold as an advancement it should be clear if this is actually the case or if it is a different approach that yields comparable and maybe (although it is not really clear which one) complimentary results. To this last point, the authors should be a bit more clear explaining what we can learn from this link-centric approach that cannot be understood for the other.

The link centric approach assumes independence between pairs of neighbors? It seems to be the case. However, to what extent this is the case? Our communications with the top people (inner circle in the Dunbar’s vision) might well be correlated.

The description of the results in the abstract (final part) is a bit cryptic

The initial motivation is a bit unclear, the “so what”m why all of this is important/relevant and how this can be applied is missing.

The order in which things are presented in the introduction is a bit confusing, authors jump between areas/topic. It should be probably re-written with more attention to the logical narration and the story telling aspect.

The authors provide some explanation on their choice of considering only the top-rank link (for both ends). However, it would interesting to see what would happen to the second, third, and also last.

The choice of three categories “(a) less than 20 years, (b) 20{39 years, and (c) at least 89

40 years” needs some more explanation.

Authors should provide some more details about Eq 1.

In line 193 authors report SVM’s accuracy of 67 % (very close to the baseline by the way) and in line 200 0.74. What is the difference there?

The authors wrote in line 439 “In contrast, the link-centric approaches

require only knowledge of the records between the ego and only one of its alters.” However some of the features they use (although they appear not to be so important) are network metrics which require more information.

The authors wrote, only in the very final part of the paper, “the problem of

prediction of user demographics from device usage data can still be considered relevant” however the reader is left wondering why this might be the case. As mentioned above much more details are needed.

6. PLOS authors have the option to publish the peer review history of their article (what does this mean?). If published, this will include your full peer review and any attached files.

Reviewer #1: No

---

## [Author Response · Author response to Decision Letter 0]

15 Nov 2019

We thank the Reviewer for his or her invaluable comments and suggestions to improve our manuscript. Here is our detailed response to the Reviewer’s report.

Reviewer comment: 

The main assumption is that differences in prediction performance across classes are associated to differences in communications patterns between such classes. It is unclear to me however to what extent this is actually proven. At the end of the day the “truth” used here comes from the outcome prediction of a quite limited set of machine learning models. In other words, how can we be sure that such differences are not introduced (even in part) by non-trivial interactions between the models and the samples? This is a key issue of machine learning approaches. In these cases, an independent theory/expectation should come to the rescue. The authors hint in several parts that some observations fit with sociological expectations and previous work however they do not provide enough details in this sense limiting their discussion with few words and citations.

Null models might be very telling. What would happen to the predictions in case the pairs would be shuffled? I am thinking about a shuffling done in the raw data (i.e. single calls) not in the aggregated metrics. The shuffling should remove the various patterns. Thus, would this imply prediction tasks more balanced across type of pairs?

Response:

The Reviewer expressed concern about the “main assumption” in our manuscript, which is that “differences in prediction performance across classes are associated to differences in communication patterns between such classes”. We would like to clarify that we did not intend to make such an assumption. However, in the previous version, it is indeed possible that a reader could get the wrong impression that the relationship between differences in prediction performance and differences in communication patterns was an assumption, rather than a result of our analysis. We have now revised the manuscript to convey our points more clearly, with the changes being in the latter part of the Introduction as well as the Results and Discussion sections. 

In order to address the Reviewer’s concern regarding “the non-trivial interactions between the models and the samples”, we emphasize in the revised manuscript the importance of the Bayes error, a model-independent quantity characterizing the intrinsic class overlap in the data. For convenience, we summarize our line of reasoning below.

The Reviewer is quite correct that using a limited number of machine learning models may not be enough to make a conclusion about the distribution of the different classes in feature space. For example, if one used only linear classifiers on a problem where the decision boundary is nonlinear, the classification accuracies would be poor but it would not reflect the separability of the data. However, for a given classification problem, there exists a statistical lower bound on the error (or equivalently, a statistical upper bound on the accuracy). This error, called the Bayes error, is nonzero if the class distributions overlap in feature space (Tumer and Ghosh, 1996). Although an exact computation of the Bayes error is practically impossible as the class distributions are unknown, estimates of the bounds of the Bayes error can still be obtained. We also note that since the Bayes error is only determined by the class distributions, it is independent of the type of classifier used in a particular classification problem.

We have made two classification tasks in our manuscript: (1) predicting whether a pair is composed of opposite-gender peers or not, and (2) whether the age of the younger user in a pair is below or at least 35. In the first case, we found from the Bayes error lower bound estimate that there is considerable inherent overlap between opposite-gender peers and other pairs in the feature space (that cannot be fixed by any classifier), and that there is an age dependence in the classification accuracy. To see whether this translates to an age dependence in the communication patterns, we looked at the results of the second classification task. Our estimate of the Bayes error lower bound indicates that there is much overlap between the two classes. However, the classifier results show that the error is mostly confined to the age groups near the age cutoff (35), while those age groups far from the cutoff are classified with high accuracy. Noting that the Bayes optimal classifier cannot perform worse than the classifiers we used, we infer that the overlap suggested by the Bayes error is confined to points near the age cutoff. Far from the cutoff, there is enough distance in feature space between the two classes to separate them, even without the use of a kernel trick (the linear SVM was the best performing classifier).

One caveat is that we represented the communication patterns as a set of 175 features derived from the call detail records. Thus, any results found should be interpreted within the framework of these 175 features. Technically speaking, what we found was an age dependence in this 175-dimensional feature space that allows us to distinguish between pairs where the age of the younger user in the pair is below 35 years (“young” group) or 35 years and older (“old” group). Reducing the call detail records to some other feature space may lead to different results, but this does not negate our findings.

The Reviewer also recommended looking at null models to address his or her aforementioned concern on how differences in prediction performance could be associated with differences in communication patterns. The Reviewer suggested in particular to shuffle the pairs in the call detail records, which should remove the various patterns, resulting in no age nor gender dependence in the data. This would essentially test how the machine learning models used would handle random inputs. As the machine learning models used are well-studied, we can trivially expect that the lack of the patterns in the data will yield a classification accuracy close to the random baseline regardless of age group. Additionally, we think that this would still not help us address the question on how to interpret our results that there are indeed differences in classification accuracy. Instead, we believe a discussion of the Bayes error in relation to class overlap and separability in feature space, as outlined above, sheds more light on this matter.

Reviewer comment:

The key element of novelty, which has been nicely stressed, is the focus on links rather than egos. However, it is unclear what is the success of approaches based on the ego. Are they better in predicting gender of pairs? This can be done predicting the gender of connected groups, I am guessing. What about age? As the authors wrote in the conclusion “the problem of prediction of user demographics from device usage data can still be considered relevant” thus it is key to understand the actual benefit of using a link centric approach is in this context. In order words, does using this approach helps? If yes, how much? Clearly the paper is not focusing 100% of improving a prediction task, but since it sold as an advancement it should be clear if this is actually the case or if it is a different approach that yields comparable and maybe (although it is not really clear which one) complimentary results. To this last point, the authors should be a bit more clear explaining what we can learn from this link-centric approach that cannot be understood for the other.

Response:

The Reviewer points out that although the novelty of the paper in focusing on the links rather than the egos has been nicely stressed, it is not clear what the success of egocentric approaches is and how the link-centric approach compares to them. We clarify that the link-centric approach we propose does not replace but complement egocentric approaches. Technically, one could use an egocentric approach to predict the age and gender of individuals and from here obtain the age and gender difference to infer the relationship between two users. However, we could not use these prediction results to gain more insights on the communication patterns between the two users because the egocentric approach requires an aggregation process over an ego’s neighbors, so pairwise information are naturally lost. Thus, link-centric approaches are best for studying the communication patterns across different relationships, which is what we set to do in this research. We have revised the manuscript to express this point more clearly, notably in the second paragraph of the Introduction and the second, third, and fourth paragraphs of the Conclusion section.

Reviewer comment:

The link centric approach assumes independence between pairs of neighbors? It seems to be the case. However, to what extent this is the case? Our communications with the top people (inner circle in the Dunbar’s vision) might well be correlated.

Response:

The Reviewer also asked about the independence between pairs of neighbors. The Reviewer has mentioned that it seems that independence between pairs of neighbors is assumed, but that if the communication patterns with the top alters (Dunbar’s inner circle) are correlated, this assumption might not be valid. Indeed, it is possible that the communication patterns among one’s own neighbors are correlated. However, we made no assumptions regarding the presence or absence of correlations between pairs, since we are only concerned with the variation in their communication patterns. Further, if these correlations exist, they may actually be the reason for the variation in communication patterns that we find. Thus, the possible level of dependence between pairs of neighbors has no effect on our analysis.

Reviewer comment:

The authors provide some explanation on their choice of considering only the top-rank link (for both ends). However, it would interesting to see what would happen to the second, third, and also last.

Response:

In our previous paper [1], we looked at the differences among communication patterns in peers, and find that mutual top-rank pairs are significantly different from those who are not. From our findings, we inferred that mutual top-rank pairs have close relationships while other pairs are more likely to have casual relationships. In the current manuscript, we focus solely on close relationships, and hence only consider mutual top-rank pairs. Although we agree that a similar study on the other type of pairs would be interesting, this is rather outside the scope of the current study.

Response to other comments by the reviewer:

Most of the Reviewer’s other comments are on the writing itself, and we have addressed this in the manuscript. The last sentence of the abstract has been revised to be more explicit. The motivation for using a link-centric approach, as well a discussion on what it can do that egocentric approaches cannot, is now emphasized in both the Introduction and Conclusion sections. The Introduction section has also been revised for better flow across the paragraphs. The choice of the three age categories was based on census data that is cited in the original manuscript; we clarify this in lines 99-100 of the marked-up copy of the revised manuscript. A more thorough discussion of the Bayes error and its estimate (eq. 1, now eq. 2) has also been included.

We also note that the 0.74 value in line 200 of the original manuscript is the precision, whereas the 67% value is the classification accuracy.

The Reviewer notes that our link-centric approach uses network metrics and thus uses the full egocentric information, and that our sentence in line 439 of the original manuscript is technically incorrect, despite the fact that the network metrics do not appear to be an important factor. We have revised this sentence (lines 529-532 in the marked-up copy) to more accurately reflect our results.

We have also altered the last paragraph of the Conclusion section to write more about how inferring demographic information from mobile phone as well as app-based communication remains relevant.

[1] M.I. Fudolig, D. Monsivais, K. Bhattacharya, H.H. Jo, K. Kaski, “Different patterns of social closeness observed in mobile phone communication”, J Comput Soc Sc 2019 https://doi.org/10.1007/s42001-019-00054-8

---

## [Editor Report · Decision Letter 1]

12 Dec 2019

Link-centric analysis of variation by demographics in mobile phone communication patterns

PONE-D-19-21537R1

Dear Dr. Fudolig,

We are pleased to inform you that your manuscript has been judged scientifically suitable for publication and will be formally accepted for publication once it complies with all outstanding technical requirements.

With kind regards,

Ginestra Bianconi

Academic Editor

PLOS ONE
---

## [Editor Report · Acceptance letter]

19 Dec 2019

PONE-D-19-21537R1 

Link-centric analysis of variation by demographics in mobile phone communication patterns 

Dear Dr. Fudolig:

I am pleased to inform you that your manuscript has been deemed suitable for publication in PLOS ONE. Congratulations! Your manuscript is now with our production department. 

With kind regards,

on behalf of

Dr. Ginestra Bianconi 

Academic Editor

PLOS ONE